# InstanT: Semi-supervised Learning with Instance-dependent Thresholds

**Muyang Li[1], Runze Wu[2], Haoyu Liu[2], Jun Yu[3], Xun Yang[3], Bo Han[4], Tongliang Liu[1]***

[1]Sydney AI Center, The University of Sydney; [2]FUXI AI Lab, NetEase;
[3]University of Science and Technology of China; [4]Hong Kong Baptist University

## Abstract

Semi-supervised learning (SSL) has been a fundamental challenge in machine learning for decades. The primary family of SSL algorithms, known as pseudo-labeling, involves assigning pseudo-labels to confident unlabeled instances and incorporating them into the training set. Therefore, the selection criteria of confident instances are crucial to the success of SSL. Recently, there has been growing interest in the development of SSL methods that use dynamic or adaptive thresholds. Yet, these methods typically apply the same threshold to all samples, or use class-dependent thresholds for instances belonging to a certain class, while neglecting instance-level information. In this paper, we propose the study of instance-dependent thresholds, which has the highest degree of freedom compared with existing methods. Specifically, we devise a novel instance-dependent threshold function for all unlabeled instances by utilizing their instance-level ambiguity and the instance-dependent error rates of pseudo-labels, so instances that are more likely to have incorrect pseudo-labels will have higher thresholds. Furthermore, we demonstrate that our instance-dependent threshold function provides a bounded probabilistic guarantee for the correctness of the pseudo-labels it assigns. Our implementation is available at `https://github.com/tmllab/2023_NeurIPS_InstanT`.

## 1 Introduction

In recent years, machine learning algorithms trained on abundant accurately labeled data have demonstrated unparalleled performance across different domains. However, in practice, it is often financially infeasible to collect reliable labels for all samples in large-scale datasets. A more practical solution is to select a subset of the data and use expert annotations to assign labels to them [39]. This scenario, where the majority of the data is unlabeled and only a portion has reliable labels, is known as semi-supervised learning (SSL). In SSL, our aim is usually to learn a classifier that has comparable performance to the one trained in a fully supervised manner. To achieve this, the majority of existing approaches adopt a training strategy named pseudo-labeling. More specifically, a model will be trained on a small set of labeled data first, and this model will then be applied to the larger unlabeled set to assign predicted pseudo-labels to them. If the model's confidence in an instance exceeds a certain threshold, then this instance along with its predicted pseudo-label will be added to the training set for this iteration. Thus by iteratively expanding the training set, if the expanded instances are indeed assigned with correct labels, then the classification error will be gradually reduced [43].

However, since the model is bound to make mistakes, the newly added training samples are not always assigned with correct labels, which generates label noise [5, 46, 44]. Under the influence of label noise, the model will gradually over-fits to noisy supervision, hence accumulating generalization error. Since the model makes predictions based on instance features, it is usually assumed that for

---

*Correspondence to Tongliang Liu (tongliang.liu@sydney.edu.au)

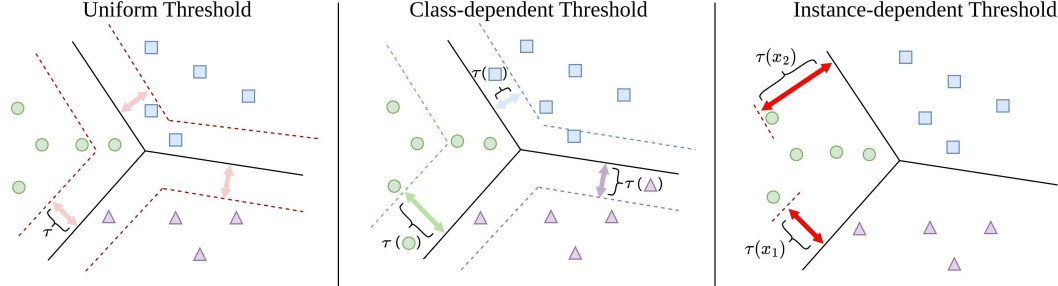

Figure 1: Illustration on the differences between uniform thresholds, class-dependent thresholds, and instance-dependent thresholds in SSL. The black solid lines are the decision boundaries generated by the classifier, and the colored dashed lines are the confidence thresholds, the colored square, triangle, and circle represent the unlabeled instances with their predicted class. We can observe that a uniform threshold does not depend on any factors, class-dependent thresholds only depend on the predicted pseudo-label, and instance-dependent thresholds depend on the features of unlabeled instances.

"hard" examples, the classifier is more likely to give incorrect predictions to them [55], making the label noise pattern **instance-dependent** [11, 45, 55].

Consequently, selecting an appropriate confidence threshold for pseudo-label assignment becomes a key factor that determines the performances of SSL methods. For predictions that are more likely to have incorrect labels, we want to assign higher thresholds to avoid selecting them prematurely. And for predictions that are more likely to be correct, we want to assign lower thresholds to encourage adding them to the training set at an earlier stage. Conventionally, such a threshold is usually determined by empirical experience in an ad-hoc manner, where only a few of the existing works consider the theoretical implication of threshold selection. Recently, increasing research focused on the investigation of dynamic or adaptive thresholds [9, 17, 41, 48, 54], where such confidence threshold is conditional on some external factors, such as the current training progress. Notably, the vast majority of those methods consider **a single threshold for all instances** and overlook their instance-level information, such as the instance-level label noise rate. As illustrated in Figure 1, the instance-dependent threshold is the most flexible among existing threshold types.

Motivated by this challenge, in this paper, we propose a new SSL algorithm named **Instan**ce-dependent **T**hresholding (**InstanT**). Since the learned model is subject to the influence of instance-dependent label error, we aim to quantify and reduce such error by estimating an instance-dependent confidence threshold function based on the potential label noise level and instance ambiguity (clean class posterior). In addition, we can derive a lower-bounded probability, for samples that satisfy the thresholds will be assigned to a correct label. From our main theorem, we show that as the training progresses, this probability lower-bound will asymptotically increase towards one, hence guaranteeing the pseudo-label quality for instances that satisfies our proposed thresholds.

We summarize our contributions as follows: (1) We propose an SSL algorithm named InstanT, which assigns thresholds to individual unlabeled samples based on the instance-dependent label noise level and prediction confidence. (2) We prove InstanT has a bounded probability to be correct, which vouch for the reliability of the pseudo-labels it assigns with a theoretical guarantee. (3) To the best of our knowledge, this is the first attempt to estimate the instance-dependent thresholds in SSL, which has the highest degree of freedom compared with existing methods. (4) Through extensive experiments, we show that our proposed method is able to surpass state-of-the-art (SOTA) SSL methods across multiple commonly used benchmark datasets.

**Related work.** As we mentioned, some existing works have already been considered to improve pseudo-labeling by leveraging "dynamic" or "adaptive" thresholds. More specifically, Dash [48] considers a monotonically decreasing dynamic threshold, based on the intuition that as training progresses, the model at a later stage will provide more reliable predictions than at early stages. Flex-Match [54] further introduced class-dependent learning status in addition to the dynamic threshold. Adsh [17] employs adaptive threshold under class imbalanced scenario by optimizing the number of pseudo-labels per class. As for adaptive thresholds, AdaMatch [9] considers using a pre-defined threshold factor multiplied by the averaged top-1 prediction confidence, making it adaptive to the

model's current confidence level. FreeMatch [41] considers a combination of adaptive global and local thresholds, which combines the training progress and class-specific information.

In addition to the aforementioned more recent methods, most of the classical methods use pre-defined fixed thresholds, which remain constant throughout the entire training process [8, 18, 19, 29, 36]. Such thresholds are usually set to be high enough to prevent the occurrence of incorrect pseudo-labels. Notably, other approaches such as reweighting [10, 22], distribution alignment [20, 31, 42], contrastive learning [24, 49], consistency regularization [1, 34] were also employed to improve SSL.

## 2 Preliminaries

### 2.1 Notation and Problem Setting

We consider the general setting from SSL, namely, we have a small group of labeled instances $X_l := \{\boldsymbol{x_1}, ..., \boldsymbol{x_n}\}$ with their corresponding labels $Y_l := \{y_1, ..., y_n\}$. And there is an unlabeled instance group $X_u := \{\boldsymbol{x_{n+1}}, ..., \boldsymbol{x_{n+m}}\}$ with their latent labels $Y_u := \{y_{n+1}, ..., y_{n+m}\}$, usually, we have $m >> n$. Furthermore, we assume that the labeled instances and unlabeled instances are independent and identically distributed (i.i.d), i.e. $X := \{X_l, X_u\} \in \mathcal{X}, Y := \{Y_l, Y_u\} \in \mathcal{Y}$.

We term the model trained with clean and pseudo-labels as $\hat{f}$, parameterized by $\theta$. Since $\hat{f}_\theta$ is bound to make mistakes, its generated pseudo-label $\hat{Y}_u$ will contain label errors. Hence the softmax predictions of $\hat{f}_\theta$ are actually approximating the noisy class posterior. We fix the notation for the real noisy class posterior as $P(\hat{Y}|X)_t$ at $t$-th iteration, whereas our approximated noise class posterior is $\hat{P}(\hat{Y}|X)_t$, which can be obtained from the softmax predictions of the model $\hat{f}_\theta$ at $t$-th iteration.

As the clean label for the unlabeled set is unknown, we instead wish to recover the *Bayes optimal label* for all the unlabeled instances, which is the label class that maximizes the clean class posterior [50, 56]. The Bayes optimal label is generated by the hypothesis that minimizes the risk within the hypothesis space, i.e. the Bayes optimal classifier, which we denote as $h^*$. And we can denote the Bayes optimal label for $\boldsymbol{x}$ as $h^*(\boldsymbol{x})$.

### 2.2 Instance-dependent Label Noise in SSL

As mentioned in the previous section, label noise is an inevitable challenge in SSL. Moreover, since $P(\hat{Y}|X)$ is dependent on $X$, we say the label noise is instance-dependent [12, 45]. Concretely, we have $P(\hat{Y}|X = \boldsymbol{x}) = T(\boldsymbol{x})P(Y|X = \boldsymbol{x})$ [32, 35], where $T(\boldsymbol{x_u})$ is the **instance-dependent transition matrix**, which models the generation of label noise. $P(Y|X = \boldsymbol{x})$ represents the latent clean class posterior for unlabeled samples. We can define the $ij$-th entry of $T(\boldsymbol{x})$ as:

$$T_{i,j}(\boldsymbol{x}) = P(\hat{Y} = j|Y = i, X = \boldsymbol{x}), \tag{1}$$

which means the $ij$-th entry of $T(\boldsymbol{x})$ models the probability that $\boldsymbol{x}$ with clean label $y = i$ will be predicted as $\hat{y} = j$.

### 2.3 Tsybakov Margin Condition

We assume our learned classifier satisfies Tsybakov Margin Condition [38], which essentially restrained the level of complexities of the classification problem by assuming the data are separable enough. It is a commonly used assumption for various Machine Learning problems [2, 6, 56] including SSL [48]. Without the loss of generality, we will directly present the multi-class Tsybakov Margin Condition [2, 56], whereas the binary case will be provided in the Appendix B.

**Assumption 1** (Multi-class Tsybakov Margin Condition). *For some finite constant $C, \alpha > 0$, and $\delta_0 \in (0, 1]$, the Tsybakov Margin Condition holds if $\forall \delta \in (0, \delta_0]$, we have*

$$P\left[P(Y = \gamma|X = \boldsymbol{x}) - P(Y = s|X = \boldsymbol{x}) \le \delta\right] \le C\delta^\alpha.$$

Where $P(Y = \gamma|X = \boldsymbol{x})$ and $P(Y = s|X = \boldsymbol{x})$ are the largest and second-largest clean class posterior probability for $\boldsymbol{x}$.

## 2.4 Quality-Quantity Trade-off in SSL

To understand the choice of pseudo-label thresholds in SSL, it is essential to consider the trade-off between the *quality* and the *quantity* of the pseudo-labels [10, 41]. Typically, a higher threshold is presumed to yield superior label quality, as it assigns pseudo-labels solely to instances where the classifier exhibits a high level of confidence. Conversely, elevating the threshold diminishes the quantity of pseudo-labels, as it leads to the exclusion of instances where the classifier lacks sufficient confidence. Hence causing the dilemma of quality-quantity trade-off.

One possible approach to address the quantity-quality trade-off in SSL is to apply a dynamic threshold, which is dependent on the training progress. This is because the evaluation metric for confident samples is usually not stationary throughout the training process of neural networks. As the model's predicted probability is subject to the level of over-fitting [16, 30], designing a thresholding function that is dependent on the training process is crucial to the success of SSL algorithms[9, 41]. Intuitively, in the early stage of the training process, where the model has just started to fitting, the threshold should not be set too high to filter out too many samples, and as the model has already well-fitted to the training data, the threshold should be elevated to combat falsely confident instances caused by confirmation bias [4].

# 3 Instance-dependent thresholds - A theoretical perspective

In this section, we focus on answering the specific research question: ***For SSL, can we derive instance-dependent pseudo-label assignment thresholds with theoretical guarantees?***

The answer to this question is affirmative. We will present our main theorem and the proof to show that for samples that satisfy our instance-dependent threshold function, their likelihood of being correct is lower-bounded.

## 3.1 Lower-bound Probability of Correctness

For a classifier trained with sufficient accurately labeled data, its prediction $\gamma$ will be made if $P(Y = \gamma | X = x) > P(Y = s | X = x)$. However, under the existence of label errors, the fidelity of the classifier can no longer be guaranteed. Recently, Zheng *et al.* [56] showed that the classifier influenced by the label noise can still have a bounded error rate. We show that in SSL tasks, where the label noise is instance-dependent, we can establish a lower-bound probability for the predictions to be correct through similar derivations.

Recall Assumption 1, we have $\gamma := \arg\max_i P(Y = i | X = x_u)$, $s := \arg\max_{i \neq k} P(Y = i | X = x_u)$, $k := \arg\max_i \hat{P}(\hat{Y} = i | X = x_u) = \hat{y}_u$. And let $\epsilon_\theta$ be the estimation error between our estimated noisy class posterior $\hat{P}(\hat{Y}|X)$ and true noisy class posterior $P(\hat{Y}|X)$. We can then formulate our main theorem:

**Theorem 1.** *Assume estimated noisy class posterior $\hat{P}(\hat{Y}|X)$ satisfies Assumption 1 with $C, \alpha > 0$, $\delta_0 \in (0, 1]$, and $\epsilon_\theta \leq \delta_0 \min_i T_{i,i}(x_u)$, we have:*

$$P\left[k = h^*(x_u), \hat{P}(\hat{Y} = k | X = x_u) \geq \tau(x_u)\right] > 1 - C[O(\epsilon_\theta)]^\alpha, \qquad (2)$$

*where $\tau(x_u)$ is the instance-dependent threshold function:*

$$\tau(x_u) = T_{k,k}(x_u)P(Y = s | X = x_u) + \sum_{i \neq k}^{|Y|} T_{i,k}(x_u)P(Y = i | X = x_u). \qquad (3)$$

The proof of Theorem 1 is provided in Appendix C. Theorem 1 substantiates that, as our model over-fits to the label error (as $\epsilon_\theta$ decreases), and the transition matrix can be successfully estimated, the probability lower-bound of the assigned pseudo-labels are correct will increase asymptotically. This guarantees the quality of the pseudo-labels $\tau(x_u)$ assigned. Note that, $\tau(x_u)$ requires the clean class posterior of unlabeled instances, which can be provably inferred upon the successful estimation of $T(x)$ and the noisy class posterior [26, 32].

To gain a deeper understanding of Theorems 1, we will try to intuitively understands the outputs of $\tau(x_u)$. Our aim is to examine the behavior of $\tau(x_u)$ in relation to the unlabeled sample $x_u$, which

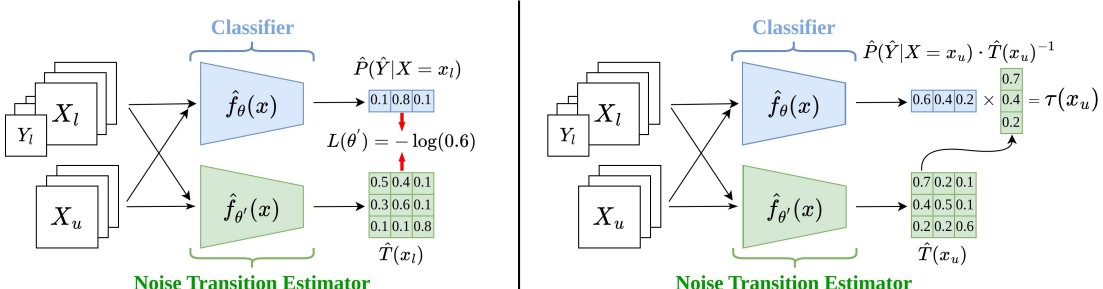

Figure 2: Overview of InstanT. **Left:** training process of transition matrix estimator $\hat{f}_{\theta'}$. For a labeled instance $x_l$, whose label is **2**, the classifier $\hat{f}_\theta$ will first generate its noisy class posterior. Then, we want the ***second row*** of the transition matrix to approximate the noisy class posterior by minimizing $L(\theta')$ from Equation 7. **Right:** The inference process of $\tau(x_u)$. When calculating $\tau(x_u)$, $x_u$ will be simultaneously passed to the classifier and transition matrix estimator. Since ***1*** is the predicted class label from the model that minimizes the forward loss (Equation 8), we will use the ***first column*** from our estimated transition matrix to modulate threshold based on Theorem 1.

has a predetermined clean class posterior. If we observe an increase in the sum of column $k$ in the transition matrix, it will result in a corresponding increase in the threshold $\tau(x_u)$. This implies that when $x_u$ is highly likely to be assigned with an incorrect pseudo-label, $\tau(x_u)$ will assign a higher threshold in order to prevent its premature inclusion of $\{x_u, \hat{y}_u\}$ in the training set. Conversely, when $x_u$ is less likely to be assigned with a incorrect pseudo-label, $\tau(x_u)$ will assign a lower threshold to encourage adding $\{x_u, \hat{y}_u\}$ to the training set.

## 3.2 Identification of Instance-dependent Transition Matrix in SSL

Theorem 1 builds upon the assumption that $T(x)$ can be successfully identified. Yet, the identification of $T(x)$ has been a long-standing issue in the field of label noise learning. Fortunately, in SSL, we argue that $T(x)$ can be provably identified under mild conditions. We will first justify our argument from a theoretical perspective based on the Theorems from Liu *et al.* [27]. Subsequently, we will also propose the empirical methods for estimating $T(x)$ from an intuitively understandable aspect in the following section.

We will start by defining the identifiability, we use $\Omega$ to represent an observation space and use $\Theta$ to represent a general parametric space. Then, for a distribution with parameter $\theta'$, such that $\theta' \in \Theta$, a model $P_{\theta'}$ on $\Omega$ can be defined [3, 50]. We further define its identifiability as:

**Definition 1** (Identifiability). *Parameter $\theta'$ is said to be identifiable if $P_{\theta'} \neq P_{\theta''}, \forall \theta' \neq \theta''$.*

In our case, $\theta'(x) := \{T(x), P(Y|X = x)\}$, i.e., $P_{\theta'}$ is a distribution defined by the transition probability and clean class posterior. To determine the identifiability of $T(x)$, we also need to define the term *informative noisy labels*. Specifically, we have:

**Definition 2** (Informative noisy labels). *For a given sample $(x, y)_i$, its noisy label $\hat{y}_i$ is said to be informative if $rank(T(x_i)) = |Y|$.*

Then, we can make the following theorem:

**Theorem 2.** *With i.i.d $(x, y)_i$ pairs, three **informative noisy labels** $\hat{y}_i$ are sufficient and necessary for the identification of $T(x_i)$.*

The Theorem 2 is based on Kruskal's Identifiability Theorem, the proof is provided in the Appendix D. From Theorem 2 and Definition 2, we can conclude that, in SSL, if we have more than three labeled samples per class, transition matrix $T(x)$ is identifiable, more discussions on this can also be found in Appendix D.

# 4 InstanT: Instance-dependent thresholds for pseudo-label assignment

## 4.1 Instance-dependent Threshold Estimation

While Theorem 1 provides a theoretical guarantee for the correctness of $\tau(\boldsymbol{x_u})$, as discussed in section 2.4, SSL algorithms can benefit from non-constant thresholds [9, 41]. Here we define the dynamic threshold $\kappa_t$ at iteration $t$ from the relative confidence threshold (RT) in AdaMatch [9]:

$$\kappa_t = \frac{1}{n}\sum_{i=1}^{n} \max_{j\in[1,...,|Y|]} \hat{P}(\hat{Y}=j|X=\boldsymbol{x_i})_t \cdot \beta, \tag{4}$$

where $n$ is the number of labeled samples, and $\beta$ is a fixed discount factor. Subsequently, we present the instance-dependent threshold function at iteration $t$:

$$\tau(\boldsymbol{x_u})_t = \min\left[1, \hat{T}_{k,k}(\boldsymbol{x_u})_t \hat{P}(Y=s|X=\boldsymbol{x_u})_t + \sum_{i\neq k}^{|Y|} \hat{T}_{i,k}\hat{P}(Y=i|X=\boldsymbol{x_u})_t + \kappa_t\right]. \tag{5}$$

Note that this estimation process requires the clean class posterior, which can be estimated or approximated using a transition matrix via importance re-weighting [26, 46] or loss correction [32, 45], details will be introduced in the following section. We also remark here that a less strict instance-dependent version of the threshold function can be used with the class-dependent transition matrix [32, 46, 52], which can lead to better empirical results under certain cases, it is also important to note that even under class-dependent transition matrix, the threshold function is still instance-dependent, as the noise class posterior are instance-dependent.

## 4.2 Modelling Instance-dependent Label Noise

Now we will introduce a piratical method for estimating $T(\boldsymbol{x})$ in SSL. Our approach involves approximating the label noise transition pattern using a Deep Neural Network. As we recall from Equation 1, $T(\boldsymbol{x})$ can be described as a mapping function $T : X \rightarrow \mathbb{R}^{|Y|\times|Y|}$, since we already know that, in general cases, $T(\boldsymbol{x})$ is identifiable in SSL, we only need to find the correct mapping function. In addition, since we have clean label $Y_l$ for $X_l$, and we also have the noisy pseudo-labels $\hat{Y}_l$ for $X_l$, this enables us to directly fit a model to approximate $T(\boldsymbol{x})$. Therefore, we want our transition matrix estimator, parameterized by $\theta'$, to approximate the label noise transition process at every iteration $t$:

$$\hat{T}_{i,j}(\boldsymbol{x_l}) = \hat{P}(\hat{Y}_l=j|Y_l=i, X_l=\boldsymbol{x_l};\theta') \approx P(\hat{Y}=j|Y=i, X=\boldsymbol{x}) = T(\boldsymbol{x}). \tag{6}$$

The above approximation holds since label noise is assumed to be i.i.d, therefore both labeled and unlabeled samples share the same noise transition process. As we can estimate the noisy class posterior for labeled samples, we can therefore collect distribution $D := \{X_l, Y_l, \hat{P}(\hat{Y}|X=\boldsymbol{x_l})\}$. Observing Equation 6, we can design following objective for $\theta'$ to minimize:

$$L_D(\theta') = -\frac{1}{n}\sum_{l=1}^{n} \hat{\boldsymbol{y_l}} \log(\boldsymbol{y_l} \cdot \hat{T}(\boldsymbol{x_l})). \tag{7}$$

Where $\boldsymbol{y_l}$ is the one-hot vector for the label of labeled samples, thus $\boldsymbol{y_l} \cdot \hat{T}(\boldsymbol{x_l})$ is equivalent to finding the class-conditional noise class posterior $\hat{P}(\hat{Y}_l=j|Y_l=i, X_l=\boldsymbol{x_l};\theta')$. $\hat{\boldsymbol{y_l}}$ is the one-hot noisy pseudo-label of $\boldsymbol{x_l}$, minimizing $L_D(\theta')$ will result $\hat{T}(\boldsymbol{x_l};\theta')$ approximate $T(\boldsymbol{x})$. As shown in Figure 2, $\hat{f}_{\theta'}$ will be trained in parallel to the main classifier and updated in every epoch.

Notably, the estimator for the transition matrix in InstanT is not constrained to one specific method, there has been a wide versatile of $T$ estimator in the field of label noise learning [25, 45, 51], we have incorporated some of them into the implementation of InstanT, indicating the good extensibility of InstanT.

### 4.3 Distribution Alignment

When the prediction of the classifier becomes imbalanced, Distribution Alignment (DA) is used to modulate the prediction so that the poorly predicted class will not be overlooked completely [7, 9, 20, 42]. This is achieved by estimating the prior of the pseudo-label $P(\hat{Y})$ during training, and setting a clean class prior $\hat{P}(Y)$, which is usually assumed to be uniformly distributed. The distribution alignment process can be described as $\hat{P}(\hat{Y} = j | X = \boldsymbol{x_u}) = \mathrm{Norm}(\hat{P}(\hat{Y} = j | X = \boldsymbol{x_u})P(Y = j)/P(\hat{Y} = j))$, where $\mathrm{Norm}$ is a total-sum scaling. Therefore, if the distribution of $\hat{Y}$ is severely imbalanced, DA will force the prediction for the minority class to be scaled up, and the prediction for the over-populated class to be scaled down.

### 4.4 Loss Correction

While the classifier trained with incorrect pseudo-labels is subject to the influence of noisy supervision, in order to satisfy Theorem 1, the estimated softmax predictions from $\hat{f}_\theta$ must approximate the clean class posterior [56]. As we have already demonstrated that the transition matrix can be identified in standard SSL setting, in this part, we will show how to infer clean class posterior from noisy predictions and transition matrix, by utilizing forward correction [32]. Specifically, let $\ell_\psi$ be a proper composite loss [33] (e.g. softmax cross-entropy loss), forward correction is defined as:

$$\overrightarrow{\ell_\psi}(\hat{\boldsymbol{y_u}}, \hat{f}_\theta(\boldsymbol{x_u})) = \ell(\hat{\boldsymbol{y_u}}, \hat{T}(\boldsymbol{x_u})^\top \psi^{-1} \hat{f}_\theta(\boldsymbol{x_u})), \tag{8}$$

where $\psi$ is an invertible link function. It has been proven that minimizing the forward loss with an accurately estimated transition matrix is equivalent to minimizing the loss defined over clean latent pseudo-labels [32]. Therefore, minimizing $\overrightarrow{\ell_\psi}$ will let the softmax prediction of classifier $\hat{f}_\theta$ approximate the clean class posterior.

### 4.5 Training with Consistency Regularization Loss

Lastly, we provide an overview of the training objective of InstanT. We employ the widely adopted consistency regularization loss [36, 47], which assumes a well-learned and robust model should generate consistent predictions for random perturbations of a given instance. Previous methods have commonly employed augmentation techniques to introduce perturbations. The training loss of classifier $\hat{f}_\theta$ consists of two parts: supervised loss $L_{D_s}(\theta)$ and unsupervised loss $L_{D_u}(\theta)$. $L_{D_s}(\theta)$ is calculated by the labeled samples, specifically, we have:

$$L_{D_s}(\theta) = \frac{1}{n} \sum_{l=1}^{n} \ell(\boldsymbol{y_l}, \hat{f}_\theta(\mathbb{W}(\boldsymbol{x_l})), \tag{9}$$

where $n$ is the number of labeled samples, $\boldsymbol{y_l}$ is the one-hot label for $\boldsymbol{x_l}$, and $\mathbb{W}$ is a weak augmentation function [36, 47], and $\ell$ is the cross-entropy loss function. Unsupervised loss $L_{D_u}(\theta)$ is calculated by the unlabeled samples $X_u$ and their predicted pseudo-labels $\hat{Y}_u$, which can be defined as:

$$L_{D_u}(\theta) = \frac{1}{m} \sum_{u=1}^{m} \mathbb{1}(\hat{P}(\hat{Y} = k | X = \mathbb{W}(\boldsymbol{x_u})) > \tau(\boldsymbol{x_u})_t) \overrightarrow{\ell_\psi}(\hat{\boldsymbol{y_u}}, \hat{f}_\theta(\mathbb{S}(\boldsymbol{x_u})), \tag{10}$$

where $m$ is the number of unlabeled instances, $\hat{\boldsymbol{y_u}}$ is the one-hot pseudo-label for $\boldsymbol{x_u}$, and $\mathbb{S}$ is a strong augmentation function [7, 14, 15]. $\mathbb{1}$ is an indicator function that filters unlabeled instances using $\tau(\boldsymbol{x_u})_t$. Combining the supervised and unsupervised loss, we then have the overall training objective:

$$L_D(\theta) = L_{D_s}(\theta) + \lambda L_{D_u}(\theta), \tag{11}$$

where $\lambda$ is used to control the influence of the unsupervised loss.

Table 1: Top-1 accuracy with pre-trained ViT. The best performance is bold and the second best performance is underlined. All results are averaged with three random seeds {0,1,2} and reported with a 95% confidence interval.

| Dataset | CIFAR-10 | | | CIFAR-100 | | | STL-10 | | |
|---|---|---|---|---|---|---|---|---|---|
| # Label | 10 | 40 | 250 | 200 | 400 | 2500 | 10 | 40 | 100 |
| PL | $37.65_{\pm3.1}$ | $88.21_{\pm5.3}$ | $95.42_{\pm0.4}$ | $63.34_{\pm2.0}$ | $73.13_{\pm0.9}$ | $84.28_{\pm0.1}$ | $30.74_{\pm6.7}$ | $57.16_{\pm4.2}$ | $73.44_{\pm1.5}$ |
| MT | $64.57_{\pm4.9}$ | $87.15_{\pm2.5}$ | $95.25_{\pm0.5}$ | $59.50_{\pm0.8}$ | $69.42_{\pm0.9}$ | $82.91_{\pm0.4}$ | $42.72_{\pm7.8}$ | $66.80_{\pm3.4}$ | $77.71_{\pm1.8}$ |
| MixMatch | $65.04_{\pm2.6}$ | $97.16_{\pm0.9}$ | $97.95_{\pm0.1}$ | $60.36_{\pm01.3}$ | $72.26_{\pm0.1}$ | $83.84_{\pm0.2}$ | $10.68_{\pm1.1}$ | $27.58_{\pm16.2}$ | $61.85_{\pm11.3}$ |
| VAT | $60.07_{\pm6.3}$ | $93.33_{\pm6.6}$ | $97.67_{\pm0.2}$ | $65.89_{\pm1.8}$ | $75.33_{\pm0.4}$ | $83.42_{\pm0.4}$ | $20.57_{\pm4.4}$ | $65.18_{\pm7.0}$ | $80.94_{\pm1.0}$ |
| UDA | $78.76_{\pm3.6}$ | $97.92_{\pm0.2}$ | $97.96_{\pm0.1}$ | $65.49_{\pm1.6}$ | $75.85_{\pm0.6}$ | $83.81_{\pm0.2}$ | $48.37_{\pm4.3}$ | $79.67_{\pm4.9}$ | $89.46_{\pm1.0}$ |
| FixMatch | $66.50_{\pm15.1}$ | $97.44_{\pm0.9}$ | $97.95_{\pm0.1}$ | $65.29_{\pm1.4}$ | $75.52_{\pm0.1}$ | $83.98_{\pm0.1}$ | $40.13_{\pm3.4}$ | $77.72_{\pm4.4}$ | $88.41_{\pm1.6}$ |
| FlexMatch | $70.54_{\pm9.6}$ | $97.78_{\pm0.3}$ | $97.88_{\pm0.2}$ | $63.76_{\pm0.9}$ | $74.01_{\pm0.5}$ | $83.72_{\pm0.2}$ | $60.63_{\pm12.9}$ | $78.17_{\pm3.7}$ | $89.54_{\pm1.3}$ |
| Dash | $74.35_{\pm4.5}$ | $96.63_{\pm2.0}$ | $97.90_{\pm0.3}$ | $63.33_{\pm0.4}$ | $74.54_{\pm0.2}$ | $84.01_{\pm0.2}$ | $41.06_{\pm4.4}$ | $78.03_{\pm3.9}$ | $89.56_{\pm2.0}$ |
| AdaMatch | $85.15_{\pm20.4}$ | $97.94_{\pm0.1}$ | $97.92_{\pm0.1}$ | $73.61_{\pm0.1}$ | $78.59_{\pm0.4}$ | $84.49_{\pm0.1}$ | $68.17_{\pm7.7}$ | $83.50_{\pm4.2}$ | $89.25_{\pm1.5}$ |
| InstanT | $87.32_{\pm10.2}$ | $97.93_{\pm0.1}$ | $98.08_{\pm0.1}$ | $74.17_{\pm0.3}$ | $78.80_{\pm0.4}$ | $84.28_{\pm0.5}$ | $69.39_{\pm7.4}$ | $85.09_{\pm2.8}$ | $89.35_{\pm1.9}$ |

## 5 Experiments

### 5.1 Experiment Setup

We use three benchmark datasets for evaluating the performances of InstanT, they are: CIFAR-10, CIFAR-100 [21], and STL-10 [13]. CIFAR-10 has 10 classes and 6,000 samples per class. CIFAR-100 has 100 classes and 600 samples per class. STL-10 has 10 classes and 500 labeled samples per class, 100,000 unlabeled instances in total. For each dataset, we set varying numbers of labeled samples to create different levels of difficulty. Following recently more challenging settings [10, 41], where the labeled samples could be extremely limited, we set the number of labeled samples per class on CIFAR-10 as $\{1, 4, 25\}$, for CIFAR-100, we set as $\{2, 4, 25\}$, for STL-10, we set as $\{1, 4, 10\}$.

To ensure fair comparison between our method and all baselines, and to allow simple reproduction of our experimental results, we implemented InstanT and conducted all experiments within USB (Unified SSL Benchmark) framework [40]. To improve the training efficiency, e.g. faster convergence, we use pre-trained ViT as the backbone model for all baselines with the same hyper-parameters in part I. We use AdamW [28] as the default optimizer, where the learning rate is set as $5e-4$ for CIFAR-10/100, $1e-4$ for STL-10 [40]. The total training iterations $K$ are set as 204,800 for all datasets. The training batch size is set as 8 for all datasets. For a more comprehensive evaluation of our proposed method, we also conduct experiments with Wide ResNet [53] trained from scratch, the detailed settings are aligned with existing works [7, 9, 36, 47, 48, 54], these results are summarized in part II. We select a range of popular baseline methods to evaluate against InstanT, which includes Pseudo-Label (PL) [23], MeanTeacher (MT) [37], VAT [29], MixMatch [8], UDA [47], FixMatch [36], Dash [48], FlexMatch [54] and AdaMatch [9]. More comprehensive setting details, including full details of hyper-parameters, can be found in Appendix A.

### 5.2 Main results

**Part I.** Our main results will be divided into two parts: part I and part II. For training efficiency, we will use a pre-trained Vision Transformer as the backbone model in part I, enabling us to compare a more diverse collection of baselines over a broader range of benchmark settings. In Part II, we will select baselines with better performance from Part I and train them from scratch for a more comprehensive evaluation.

Table 2: Running time on STL-10(40)[2].

| Method | s/epoch |
|---|---|
| FixMatch | 30.1 |
| AdaMatch | 30.1 |
| InstanT | 31.1 |

Experimental results from part I are summarized in Table 1, where we can observe that, overall, InstanT achieves the best performances in most of the settings. More specifically, in the setting where label amount is extremely limited, InstanT exhibits the most significant improvement over all datasets, including an average 2.17% increase in accuracy on CIFAR-10 (10) over SOTA. While the identifiability of the transition matrix cannot be guaranteed with extremely limited labels, this could be remedied by including highly confident instances and their pseudo-labels to the labeled set and using them to better estimate the transition matrix. However, as the number of labeled samples increases, the

---

[2]Running time is tested on NVIDIA RTX 4090 GPUs.

Table 3: Top-1 accuracy with WRN-28. The best performance is bold and the second best performance is underlined. All results are averaged with three random seeds {0,1,2} and reported with a 95% confidence interval.

| Dataset | CIFAR-10 | | | CIFAR-100 | | |
|---|---|---|---|---|---|---|
| # Label | 40 | 250 | 4000 | 400 | 2500 | 10000 |
| UDA | $91.60_{\pm 1.5}$ | $94.44_{\pm 0.3}$ | $95.55_{\pm 0.0}$ | $40.60_{\pm 1.8}$ | $64.79_{\pm 0.8}$ | $72.13_{\pm 0.2}$ |
| FixMatch | $91.45_{\pm 1.7}$ | $\mathbf{94.87_{\pm 0.1}}$ | $95.51_{\pm 0.1}$ | $45.08_{\pm 3.4}$ | $65.63_{\pm 0.4}$ | $71.65_{\pm 0.3}$ |
| FlexMatch | $94.53_{\pm 0.4}$ | $\underline{94.85_{\pm 0.1}}$ | $\mathbf{95.62_{\pm 0.1}}$ | $47.81_{\pm 1.4}$ | $66.11_{\pm 0.4}$ | $\mathbf{72.48_{\pm 0.2}}$ |
| Dash | $84.67_{\pm 4.3}$ | $94.78_{\pm 0.3}$ | $95.54_{\pm 0.1}$ | $45.26_{\pm 2.6}$ | $65.51_{\pm 0.1}$ | $72.10_{\pm 0.3}$ |
| AdaMatch | $94.64_{\pm 0.0}$ | $94.76_{\pm 0.1}$ | $95.46_{\pm 0.1}$ | $\underline{52.02_{\pm 1.7}}$ | $66.36_{\pm 0.7}$ | $\underline{72.32_{\pm 0.2}}$ |
| InstanT | $\mathbf{94.83_{\pm 0.1}}$ | $94.72_{\pm 0.2}$ | $\underline{95.57_{\pm 0.0}}$ | $\mathbf{53.94_{\pm 1.8}}$ | $\mathbf{67.09_{\pm 0.0}}$ | $72.30_{\pm 0.4}$ |

dominance of InstanT becomes less pronounced. We hypothesize that this can be attributed to the classifier $\hat{f}_\theta$ performing better with more labeled samples, resulting in fewer label errors. Since InstanT primarily aims to increase the threshold for instances likely to have noisy pseudo-labels, the impact of InstanT becomes less significant in the presence of reduced label noise. Moreover, comparing the performance gap between InstanT and its closest counterpart, AdaMatch, it becomes apparent that InstanT consistently outperforms AdaMatch in nearly all cases. This observation underscores the non-trivial improvement and contribution brought forth by InstanT.

**Part II.** In this part, we will present the experiment results of part II, which are summarized in Table 3. Specifically, we will compare InstanT against other popular baseline methods on CIFAR-10/100 trained from scratch. For all methods, we use WRN-28-2 as default backbone model, trained with $2^{20}$ iterations. We use SGD as the default optimizer, with momentum set as 0.9, initial learning rate as 0.03 and a cosine learning rate scheduler.

Observing from experiment results, we can summarize similar patterns as the results from part I. Overall, InstanT brings most significant performance increases when the number of labeled samples are limited, e.g., when there is only 4 labeled samples per-class. Notably, for CIFAR-100 (400), InstanT exhibits an increase in accuracy for nearly 2%, which can be view as a significant improvement over SOTA baseline. While on other cases such as CIFAR-10 (250) and CIFAR-100 (10000), InstanT outperformed by other baseline methods, we emphasize that no single baseline consistently outperforms InstanT in all cases.

Another question of interest is whether InstanT will be as efficient as other methods in training time. While learning transition matrix estimator does sacrifice time and space complexity to some extent, we show that InstanT can still maintain high efficiency. As shown in Table 2, compared with FixMatch and AdaMatch, InstanT almost does not introduce further significant computational burden.

To gain a better understanding of the key differences between InstanT and existing methods, we now focus on the visualizations of InstanT compared to representative baselines. Specifically, we will display the classification accuracy, unlabeled sample utilization ratio, and pseudo-label accuracy of InstanT, FreeMatch [41], AdaMatch [9], and FixMatch [36] in Figure 3. These models are trained from scratch on CIFAR-100 with 400 labeled samples, following commonly used settings [36, 41].

Examining the classification accuracy in Figure 3(a), we observe that while InstanT does not converge as rapidly as FreeMatch, it ultimately achieves superior accuracy in the later stages of training. This is attributed to InstanT's ability to find a better balance between the quantity and quality of pseudo-labels. As depicted in Figure 3(b) and 3(c), although InstanT utilizes fewer unlabeled samples compared to FreeMatch, it demonstrates significantly improved accuracy on pseudo-labels. On the other hand, AdaMatch exhibits similar pseudo-label accuracy to InstanT but utilizes a smaller proportion of unlabeled instances. The key distinction lies in the instance-dependent threshold function employed by InstanT, which allows us to set a lower $\kappa_t$ for InstanT compared to AdaMatch, as InstanT mitigates the negative impact by increasing the instance-dependent thresholds for instances more prone to label noise, whereas AdaMatch will incorporate too many instances with incorrect pseudo-labels.

### 5.3 Ablation study

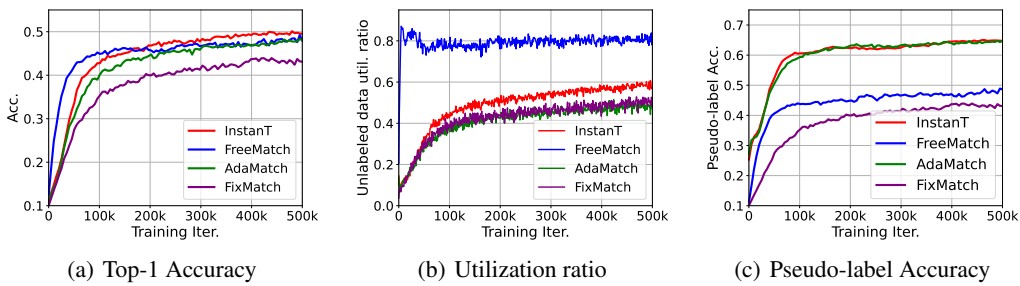

| (a) Top-1 Accuracy | (b) Utilization ratio | (c) Pseudo-label Accuracy |

Figure 3: Results from InstanT and selected baselines, trained from scratch on CIFAR-100(400).

In this section, we will be verifying the effects of each component of InstanT, specifically, we will focus on evaluating the parts that we adapted from existing works, and determine how many performance improvements are attributed to our instance-dependent thresholds. Results of this ablation study are summarized in Table 4, where all experiments are averaged over three random seeds and ran on STL-10(40) using the setting of part I. First, we observe that InstanT-I removed both Distribution Alignment (DA) and relative thresholds (RT), which makes it equivalent

Table 4: Ablation study on STL-10(40).

| Method | RT | DA | Acc. |
|---|---|---|---|
| FixMatch | ✗ | ✗ | $77.72_{\pm 4.4}$ |
| AdaMatch | ✓ | ✓ | $83.50_{\pm 4.2}$ |
| InstanT-I | ✗ | ✗ | $79.92_{\pm 5.8}$ |
| InstanT-II | ✓ | ✗ | $81.49_{\pm 4.9}$ |
| InstanT-III | ✗ | ✓ | $82.97_{\pm 2.7}$ |
| InstanT | ✓ | ✓ | $\mathbf{85.09}_{\pm \mathbf{2.8}}$ |

to adding $\tau(\boldsymbol{x_u})$ on top of the fixed threshold of FixMatch. Notably, this also brings a significant improvement over FixMatch for over 2%, which further verifies the effectiveness of our proposed instance-dependent thresholds. Simply removing RT or DA will also cause a performance drop, which is aligned with the results from AdaMatch [9].

## 6 Conclusion

In this paper, we introduce a novel approach to thresholding techniques in SSL called instance-dependent confidence threshold. This approach offers the highest level of flexibility among existing methods, providing significant potential for further advancements. We then present InstanT, a theoretically guided method designed under the concept of instance-dependent threshold. InstanT assigns unique confidence thresholds to each unlabeled instance, considering their individual likelihood of having incorrect labels. Additionally, we demonstrate that our proposed method ensures a bounded probability of assigning correct pseudo-labels, a characteristic rarely offered by existing SSL approaches. Through extensive experiments, we demonstrate the competitive performance of InstanT when compared to SOTA baselines.

## 7 Acknowledgement

Tongliang Liu is partially supported by the following Australian Research Council projects: FT220100318, DP220102121, LP220100527, LP220200949, and IC190100031. Muyang Li, Runze Wu and Haoyu Liu are supported by NetEase Youling Crowdsourcing Platform. This research was undertaken with the assistance of resources from the National Computational Infrastructure (NCI Australia), an NCRIS enabled capability supported by the Australian Government. The authors acknowledge the technical assistance provided by the Sydney Informatics Hub, a Core Research Facility of the University of Sydney.

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

## A Comprehensive settings

For all methods, we apply temperature scaling for probability calibration by default, and the scaling factor is set as 0.5. For FixMatch, Dash, FlexMatch and AdaMatch, $\tau$ is set as 0.95. For UDA, $\tau$ is set as 0.8, temperature scaling factor is set as 0.4, according to the recommendation of the original paper. For Dash, $\gamma$ is set as 1.27, $C$ is set as 1.0001, $\rho$ is set as 0.05, warm-up iteration is set as 5120 iterations. For InstanT, base $\tau$ is set as 0.9 by default.

## B Tsybakov Margin Condition (Binary)

We present the binary case of Tsybakov Margin Condition here. It can be defined as:

**Assumption 2** (Binary Tsybakov Margin Condition). *For some finite constant $C, \alpha > 0$, and $\delta_0 \in \left(0, \frac{1}{2}\right]$, the Tsybakov Margin Condition holds if $\forall \delta \in (0, \delta_0]$, we have*

$$P\left[\left|P(Y|X = \boldsymbol{x}) - \frac{1}{2}\right| \le \delta\right] \le C\delta^{\alpha}.$$

*Through similar derivations, we can provide our main theorem in binary case:*

**Theorem 3.** *Assume estimated noisy class posterior $\hat{P}(\hat{Y}|X)$ satisfies Assumption 2 with $C, \alpha > 0$, $\delta_0 \in \left(0, \frac{1}{2}\right]$, and $\epsilon_\theta \le \delta_0(1 - T_{1,0(\boldsymbol{x}) - T_{0,1}(\boldsymbol{x})})$, we have:*

$$P\left[k = h^*(\boldsymbol{x_u}), \hat{P}(\hat{Y} = k|X = \boldsymbol{x_u}) \ge \tau(\boldsymbol{x_u})\right] > 1 - C[O(\epsilon_\theta)]^{\alpha}, \tag{12}$$

*where $\tau(\boldsymbol{x_u})$ is the instance-dependent threshold function under binary case:*

$$\tau(\boldsymbol{x_u}) = T_{k,k}(\boldsymbol{x_u})P(Y = s|X = \boldsymbol{x_u}) + \sum_{i \ne k}^{|Y|} T_{i,k}(\boldsymbol{x_u})P(Y = i|X = \boldsymbol{x_u}). \tag{13}$$

## C Proof of Theorem 1

**Definition 3.** *For multi-class classification, the relationship between noisy class posterior $P(\hat{Y}|X)$ and clean class posterior $P(Y|X)$ can be concluded as:*

$$P(\hat{Y} = j|X = x) = \sum_{i=1}^{|Y|} T_{i,j}(X = \boldsymbol{x})P(Y = i|X = x). \tag{14}$$

**Proof of Theorem 1**

For simplicity, we will denote the clean class posterior $P(Y = k|X = \boldsymbol{x})$ as $\eta_k(\boldsymbol{x})$, noise class posterior $P(\hat{Y} = k|X = \boldsymbol{x})$ as $\eta_{\hat{k}}(\boldsymbol{x})$, estimated noise class posterior $\hat{P}(\hat{Y} = k|X = \boldsymbol{x})$ as $\hat{\eta}_{\hat{k}}(\boldsymbol{x})$.

*Proof.*

$$P\left[k = h^*(\boldsymbol{x_u}), \hat{\eta}_{\hat{k}}(\boldsymbol{x_u}) \ge \tau(\boldsymbol{x_u})\right] = 1 - P\left[k = h^*(\boldsymbol{x_u}), \hat{\eta}_{\hat{k}}(\boldsymbol{x_u}) < \tau(\boldsymbol{x_u})\right]$$
$$= 1 - P\left[\eta_k(\boldsymbol{x_u}) \ge \eta_s(\boldsymbol{x_u}), \hat{\eta}_{\hat{k}}(\boldsymbol{x_u}) < \tau(\boldsymbol{x_u})\right]$$

Based on Definition 3, we can substitute $\eta_{\hat{k}}(\boldsymbol{x_u})$ with $\sum_{i=1}^{|Y|} T_{i,k}(\boldsymbol{x_u})\eta_i(\boldsymbol{x_u})$, and since $\|\eta_{\hat{k}} - \hat{\eta}_{\hat{k}}\|_\infty \leq \epsilon_\theta$ ($\epsilon$-close), we can continue with:

$$\leq 1 - P\left[\eta_k(\boldsymbol{x_u}) \geq \eta_s(\boldsymbol{x_u}), \sum_{i=1}^{|Y|} T_{i,k}(\boldsymbol{x_u})\eta_i(\boldsymbol{x_u}) < \tau(\boldsymbol{x_u}) + \epsilon_\theta\right]$$

$$= 1 - P\left[\eta_k(\boldsymbol{x_u}) \geq \eta_s(\boldsymbol{x_u}), \eta_k(\boldsymbol{x_u}) < \frac{\tau(\boldsymbol{x_u}) - \sum_{i \neq k}^{|Y|} T_{i,k}(\boldsymbol{x_u})\eta_i(\boldsymbol{x_u}) + \epsilon_\theta}{T_{k,k}(\boldsymbol{x_u})}\right]$$

$$= 1 - P\left[\eta_s(\boldsymbol{x_u}) \leq \eta_k(\boldsymbol{x_u}) < \frac{\tau(\boldsymbol{x_u}) - \sum_{i \neq k}^{|Y|} T_{i,k}(\boldsymbol{x_u})\eta_i(\boldsymbol{x_u})}{T_{k,k}(\boldsymbol{x_u})} + \frac{\epsilon_\theta}{T_{k,k}(\boldsymbol{x_u})}\right]$$

$$= 1 - P\left[\eta_s(\boldsymbol{x_u}) \leq \eta_k(\boldsymbol{x_u}) < \eta_s(\boldsymbol{x_u}) + \frac{\epsilon_\theta}{T_{k,k}(\boldsymbol{x_u})}\right]$$

Note that, since $\epsilon_\theta \leq t_0 \min_k T_{k,k}(\boldsymbol{x_u})$, the Multi-class Tsybakov Condition can be applied here:

$$P\left[0 \leq \eta_k(\boldsymbol{x_u}) - \eta_s(\boldsymbol{x_u}) < \frac{\epsilon_\theta}{T_{k,k}(\boldsymbol{x_u})}\right] \leq C\left[\frac{\epsilon_\theta}{T_{k,k}(\boldsymbol{x_u})}\right]^\alpha$$

Which gives us:

$$1 - P\left[0 \leq \eta_k(\boldsymbol{x_u}) - \eta_s(\boldsymbol{x_u}) < \frac{\epsilon_\theta}{T_{k,k}(\boldsymbol{x_u})}\right] > 1 - C\left[\frac{\epsilon_\theta}{T_{k,k}(\boldsymbol{x_u})}\right]^\alpha$$

Hence proved $P\left[k = h^*(\boldsymbol{x_u}), \hat{\eta}_{\hat{k}}(\boldsymbol{x_u}) \geq \tau(\boldsymbol{x_u})\right] > 1 - C[O(\epsilon_\theta)]^\alpha$. $\qquad\square$

## D Proof of Theorem 2

In this section, we provide the proof from Liu *et al.* for the necessary and sufficient condition of the identification of instance-dependent transition matrix $T(\boldsymbol{x})$. First, we provide the Kruskal's Identifiability Theorem (or Kruskal rank Theorem):

**Theorem 4.** *The parameters $M_i, i = 1, ..., p$ are identifiable, up to label permutation, if*

$$\sum_{i=1}^{p} (M_i) \geq 2|Y| + p - 1 \tag{15}$$

*Proof.* Then we follow the proof from Liu *et al.* to first prove the sufficiency. Since we cannot observe the true label for unlabeled instances $X_u$, we denote them as hidden variable $Z$ for simplicity. $P(Z = i)$ is the prior for the latent true label.

Each $\hat{Y} = i, \forall i = 1, ..., p$ corresponds to the observation $O_i$. $|\hat{Y}|$ is the cardinality of the pseudo-label space. Without loss of generality, we will fix the number of classes as three during the analysis.

Each $\hat{Y}_i$ corresponds to an observation matrix $M_i$:

$$M_i[j, k] = P(O_i = k|Z = j) = P(\hat{Y}_i = k|Y = j, X = \boldsymbol{x})$$

Therefore, by definition of $M_1, M_2, M_3$ and $T(\boldsymbol{x})$, they all equal to $T(\boldsymbol{x})$: $M_i \equiv T(X), i = 1, 2, 3$. When $T(\boldsymbol{x})$ has full rank, we know immediately that all rows in $M_1, M_2, M_3$ are independent. Therefore, the Kruskal ranks satisfy

$$(M_1) = (M_2) = (M_3) = |\hat{Y}|$$

Checking the condition in Theorem 4, we easily verify

$$(M_1) + (M_2) + (M_3) = 3|\hat{Y}| \geq 2|\hat{Y}| + 2$$

Where Theorem 4 proves the sufficiency for three labeled samples per-class is sufficient for the identification of $T(\boldsymbol{x})$

Subsequently, Liu *et al.* also provides the proof for necessity, without the loss of generality, the analysis for this part will be conducted in binary case only. More specifically, the label error generation in SSL can be described by a binary transition matrix:

$$T(\boldsymbol{x}) = \begin{bmatrix} 1 - e_-(\boldsymbol{x}) & e_-(\boldsymbol{x}) \\ e_+(\boldsymbol{x}) & 1 - e_+(\boldsymbol{x}) \end{bmatrix}$$

In order to prove necessity, we must show that less than three informative labels per-class will not give us a unique solution for $T(\boldsymbol{x})$. If we were given two unlabeled instances with their pseudo-label $\hat{Y}_1$ and $\hat{Y}_2$. Specifically, with two unlabeled instance and pseudo-label pairs, we can conclude the following probabilities can derive all potential probabilities we can possibly get from them:

Posterior: $P(\hat{Y}_1 = +1|X)$

Positive Consensus: $P(\hat{Y}_1 = \hat{Y}_2 = +1|X)$

Negative Consensus: $P(\hat{Y}_1 = \hat{Y}_2 = -1|X)$

This is because other probability distributions are able to be derived from these three quantities:

$$P(\hat{Y}_1 = -1|X) = 1 - P(\hat{Y}_1 = +1|X) \,,$$
$$P(\hat{Y}_1 = +1, \hat{Y}_2 = -1|X) = P(\hat{Y}_1 = +1|X) - P(\hat{Y}_1 = \hat{Y}_2 = +1|X) \,,$$
$$P(\hat{Y}_1 = -1, \hat{Y}_2 = +1|X) = P(\hat{Y}_2 = +1|X) - P(\hat{Y}_1 = \hat{Y}_2 = +1|X) \,.$$

But $P(\hat{Y}_2 = +1|X) = P(\hat{Y}_1 = +1|X)$, since noisy labels are i.i.d. The above three quantities led to three equations that depend on $e_+, e_-$. This gives us the following system of equations:

$$P(\tilde{Y} = +1|X) = P(Y = +1) \cdot (1 - e_+) + (1 - P(Y = +1)) \cdot e_-$$
$$P(\hat{Y}_1 = \hat{Y}_2 = +1|X) = P(Y = +1) \cdot (1 - e_+)^2 + (1 - P(Y = +1)) \cdot e_-^2$$
$$P(\hat{Y}_1 = \hat{Y}_2 = -1|X) = P(Y = +1) \cdot e_+^2 + (1 - P(Y = +1)) \cdot (1 - e_-)^2$$

Where:

$$P(\hat{Y}_1 = \hat{Y}_2 = +1|X)$$
$$= P(\hat{Y}_1 = \hat{Y}_2 = +1, Y = +1|X)$$
$$\quad + P(\hat{Y}_1 = \hat{Y}_2 = +1, Y = -1|X)$$
$$= P(\hat{Y}_1 = \hat{Y}_2 = +1|Y = +1, X) \cdot P(Y = +1|X)$$
$$\quad + P(\hat{Y}_1 = \hat{Y}_2 = +1|Y = -1, X) \cdot P(Y = -1|X)$$
$$= P(Y = +1) \cdot (1 - e_+)^2 + (1 - P(Y = +1)) \cdot e_-^2$$

Given $Y, \hat{Y}_1, \hat{Y}_2$ are conditional independent, which gives us:

$$P(\hat{Y}_1 = \hat{Y}_2 = +1|Y = +1, X) =$$
$$P(\hat{Y}_1 = +1|Y = +1, X)(\hat{Y}_2 = +1|Y = +1, X)$$
$$P(\hat{Y}_1 = \hat{Y}_2 = +1|Y = -1, X) =$$
$$P(\hat{Y}_1 = +1|Y = -1, X)(\hat{Y}_2 = +1|Y = -1, X)$$

Conversely, for $P(\hat{Y}_1 = \hat{Y}_2 = -1|X)$, the derivation can follow similar procedure. We can now assert that $e_+, e_-$ are not identifiable. Where a simple counter example can show that two sets of unique $e_+, e_-$ both satisfies above derivation:

- $P(Y = +1) = 0.7$, $e_+ = 0.2$, $e_- = 0.2$
- $P(Y = +1) = 0.8$, $e_+ = 0.242$, $e_- = 0.07$

Which proves that two informative noisy labels are insufficient to identify $T(\boldsymbol{x_u})$. $\qquad\square$

