# OpenReview forum: "InstanT: Semi-supervised Learning with Instance-dependent Thresholds"
_NeurIPS.cc/2023/Conference — NeurIPS 2023 poster_

### Official Review · Reviewer_4nX9 · 2023-07-04

**Soundness:** 4 excellent
**Presentation:** 4 excellent
**Contribution:** 4 excellent
**Rating:** 7
**Confidence:** 4

**Summary:**

This paper introduces a new thresholding methodology for Semi-Suervised Learning. This paper proposes InstanT, which uses an instance-dependent threshold for each unlabeled data (Fig1). This algorithm shows the improvement on multiple semi-supervised benchmark datasets.

**Strengths:**

Pros.
- This paper is well-written and easy to follow.
- This paper is well-motivated (Fig 1).
- This paper proposes InstanT, and it is very well derived.
- This paper shows extensive experiments (Especially it seems to train all other semi-supervised learning baselines using ViT models) and shows the effectiveness of this method.

**Weaknesses:**

Cons.
- (Minor) This paper uses a pre-trained ViT model. In pre-training, I think the classes in Cifar10/Cifar100 could be the subset of the pre-training dataset. If These classes are already trained, they could be already well clustered and easy to train with a very small labeled dataset. (It also shows the performance improvement in the original training setting in the Appendix)

- (Minor) It would be great if it reports the supervised learning results (full labeled dataset)

- (Major) This paper evaluates their algorithm in very small datasets (Cifar10/Cifar100/STL-10). Therefore, I don't know how it robustly works in large datasets such as ImageNet. Nowadays, most of the papers evaluate their algorithms on ImageNet. (In this case, it seems to train the model from scratch. (not using pre-trained ViT model)


**Questions:**

Please see the weaknesses.

**Limitations:**

Please see the weaknesses.

---

> ### Author Rebuttal · Authors · 2023-08-10
>
> Thank you for acknowledging our contributions. We are grateful for your constructive feedbacks and positive remarks about our work. We hope we can address you questions and concerns:
>
> > **W1: Are the classes from CIFAR10/100 a subset of pre-trained dataset?**
>
> Thank you for this very insightful concern, the pre-trained dataset is ImageNet-1k, and there indeed are some class overlaps between CIFAR-100 and ImageNet-1k. However, we would like to raise several points proving our results are indeed convincing:
>
>  - the distribution of features are still notably different (different reslution level, different feature space) between CIFAR and ImageNet;
>
>  - even though the pre-trained ViT might have some prior knowledge to CIFAR10/100, this does not harm using pre-trained ViT as the backbone model. Because all the baseline methods are running on the same backbone, and the fact that InstanT can still obtain better performances showcase its improvement is valid and non-trivial;
>
>  - lastly, as you mentioned, we can also observe performance improvements on train-from-scratch cases in the Appendix. We also supplement you with some our newest results comparing with newly added baselines when train from scratch:
>
> | Methods | CIFAR-10(10) | CIFAR-10(40) | CIFAR-10(250) | CIFAR-100(400) | CIFAR-100(2500) |
> | ------ | ------ | ------ | ------ | ------ | ------ |
> | SoftMatch | 0.7557 | 0.9464 | 0.9517 | 0.5057 | 0.6622 |
> | FreeMatch | 0.9193 | **0.9512** | 0.9506 | 0.4920 | 0.6659 |
> | InstanT | **0.9250** | 0.9510 | **0.9525** | **0.5217** | **0.6709** |
>
> Table 5-1
>
> Based on the above justifications, we hope you find our evaluations to be sensible and our improvements are significant.
>
> > **W2: Did not include fully-supervised results.**
>
> Thank you for this valuable suggestion, we have supplement the sully-supervised results and will update them into our paper, we also present them here to address your concern:
>
> pre-trained ViT results, settings are aligned with Table 1 of the paper.
>
> | CIFAR-10 | CIFAR-100 | STL-10 |
> | ------ | ------ | ------ |
> | 0.991 ±0.00 | 0.9152 ±0.00 | 0.8100 ±0.03 |
>
> Table 5-2
>
> We can observe that STL-10 exhibts results that are worse than SSL baselines, this is bacause unlike CIFAR datasets, the vast majority of STL-10 do not come with groun-truth labels, so fully-supervised models trained on STL-10 can only access to a small proportion of labels.
>
> **> W3: Did not include large-scale datasets.**
>
> Per your suggestion, we have implemented experiments on ImageNet-100. And, as you have pointed out, we did not use pre-trained ViT, since that will be ill-posed. Results are trained form scratch with a ResNet-50 for 500,000 iterations on ImageNet-100 using random seed {0}.
>
> | Methods | Top-1 Accuracy | F-1 Score |
> | ------ | ------ | ------ |
> | FixMatch | 0.6624 | 0.6559 |
> | AdaMatch | 0.6860 | 0.6822 |
> | FreeMatch | 0.6578 | 0.6529 |
> | InstanT | **0.6994** | **0.6972** |
>
> Table 5-3
>
> From Table 5-3, we can observe that InstanT maintains a robust performance on real-world large-scale datasets, surpassing several SOTA baselines. We will supplement the more comprehensive results of this experiment with the main paper.
>
> Once again, we would like to thank the reviewer for your positive remarks and spending your valuable time reviewing our paper, we welcome any new questions & suggestions the reviewer might have after rebuttal.

---

> > ### Comment · Reviewer_4nX9 · 2023-08-16
> >
> > All of my concerns are addressed. So, I changed my score.

---

### Official Review · Reviewer_kWe6 · 2023-07-05

**Soundness:** 2 fair
**Presentation:** 3 good
**Contribution:** 2 fair
**Rating:** 5
**Confidence:** 3

**Summary:**

This paper focuses on semi-supervised learning (SSL) and proposes the study of instance-dependent thresholds to make incorrect pseudo-labels have higher thresholds.

**Strengths:**

This paper studies an important problem in SSL, and tries to propose a dynamic and adaptive threshold to guarantee the pseudo-label quality for instances. Also, it gives a theoretical analysis on estimating the threshold. The experiments are sufficient and the results surpasses SOTA. In the results, it is obvious to see that the algorithm has a considerable improvement in Top-1 accuracy. The ablation study shows that every part of this algorithm is useful.

**Weaknesses:**

Theorem 2 is proved with Bayes rule with Assumption 2. But if the dynamic value k_t > 0, Theorem 2 does not hold. It is incorrect P(X>=a+b)>=P(X>=a) for any b>0. I think there may be some mistakes in Assumption 2 and Theorem 2. The theorems and analyses based on Assumption 2 may be wrong.
In section 4.1 the authors do not give any theoretical reasons for the definition of k_t and \tau(x_u). Whether k_t satisfies Assumption 2 or not?
In Equation 8, the authors use an approximation \hat{T}(x) to estimate T(x). But the analysis is based on the real value of T(x). If the label noise is large or the number of samples is not enough, the error between the real value of T(x) and the approximation \hat{T}(x) cannot be ignored. Some analysis should be provided here.


**Questions:**

As mentioned in the part of weaknesses, the authors may make a mistake in Theorem 2, which makes that the paper is not convincing. The theoretical analysis part of the paper should be considered more seriously.
The dynamic threshold is determined by the transition matrix T(x). But in practice it is impossible to estimate it without any bias. The bias may influence on the final result. Is it possible to provide an error bound in the analysis?

---

> ### Author Rebuttal · Authors · 2023-08-10
>
> We thank the reviewer for carefully going over our proof and pose challenges to them, we strongly agree that the theoritical part of the paper should be rigorous, and believe that your questions and suggestions will further refine our paper.
>
>  > **W1: Proof regarding to Theorem 2.**
>
> For simplicity, let's denote event $k=h^{*}(\bf{x_u})$ as event A, $\hat{P}(\hat{Y}=k|X=\bf{x_u}) \geq \tau$ as event B, $\hat{P}(\hat{Y}=k|X=\bf{x_u}) \geq \tau + \kappa$ as event C, without the loss of generality, iteration $t$ will be omitted.
>
> First of all, you're correct, for any $\kappa \geq 0$, $P(B) \geq P(C)$ holds. However, the relationship between the joint proability of $P(A,B)$ and $P(A,C)$ are not known, since event A,B,C are not independent. Therefore, we must first decompose them into conditional probabilities and making assumptions on their relationship (Assumption 2).
>
> To understand Assumption 2 with a concrete example, consider this SSL dataset with total of 1000 unlabeled samples. With a non-dynamic fixed threshold $\tau$, the prediction confidence of all unlabeled samples surpass this threshold, P(B) = 1, only 500 of samples are assigned with correct thresholds, P(A|B) = 0.5. If we increase the threshold with some non-zero constant $\kappa$, now only 600 samples surpass this threshold, and among them, there is still 500 samples assigned with correct threshold, P(C) = 0.6, P(A|C) = 5/6. Note P(A,B) = 0.5, and P(A,C) = 0.5 as well, which satisfies Assumption 2.
>
> Nevertheless, we are also putting Assumption 2 under the microscope, examining its rigor and necessity, it's possible that we might need to modify Assumption 2 and make more rigorous justifications.
>
> > **W2: Does $\kappa_t$ in section 4.1 satisfies Assumption 2?**
>
> Thanks for pointing this question out. $\kappa_t$ is assumed to satisfy Assumption 2 throughout the paper, we will explicitly clarify this point in the final version of the paper.
>
> > **W3: Didn't account the estimation error of $T(x)$**
>
> Thank you for this highly constructive suggestion. We have improved our proof and including the estimation error of $T(x)$. Our improved proof can be found in the submitted pdf file in the genral response at the top.
>
> As for the estimation error bound of $T(x)$, it has been extensively studied and is been continuously improved, recent paper indicting even with a small number of samples, the estimation error can still be managed to reduce to a small level. Lastly, and more importantly, emprical results on challenging cases (where the estimation error of $T(x)$ is larger), still suggests our method can obtain a more robust performance than SOTA baselines.
>
> Furthermore, we wish to underscore our central contribution, which lies in the introduction and estimation of instance-level label errors. This forms the foundation for deriving instance-dependent thresholds in SSL. We view our work as an initial exploration into the possibility of instance-dependent thresholds in SSL.
>
> Once again, we would like to express our appreciation to the reviewer for carefully checking our proof and results. We welcome any new questions & suggestions you might still holds after the rebuttal.

---

> ### Author Response · Authors · 2023-08-16
> **Invitation to the rolling discussion**
>
> Dear reviewer kWe6, we hope our rebuttal has satisfactorily addressed your concerns. We are looking forward to discussing with you during the rolling discussion phase, for we genuinely feel that your valuable insights can undoubtedly further enhance our paper.

---

### Official Review · Reviewer_Cia8 · 2023-07-05

**Soundness:** 3 good
**Presentation:** 3 good
**Contribution:** 3 good
**Rating:** 7
**Confidence:** 4

**Summary:**

This paper presents a new approach for selecting confident examples called InstanT, which uses instance-dependent thresholds for assigning pseudo-labels to unlabeled data. Unlike existing methods that apply the same threshold to all samples, InstanT considers the instance-level ambiguity and error rates of pseudo-labels, assigning higher thresholds to instances more likely to have incorrect pseudo-labels. The paper demonstrates that this approach provides a probabilistic guarantee for the correctness of the assigned pseudo-labels. This innovative method may offer a new perspective on SSL.

**Strengths:**

1.The paper finds a significant and challenging problem in semi-supervised learning (SSL) that has not been adequately tackled by existing methods. Traditional SSL methods typically use a single loss threshold to select confident examples, implicitly assuming that examples with the same loss have the same likelihood of pseudo-label correctness. However, this assumption does not always hold, as there can be hard but confident examples that have larger loss values but correct pseudo-labels.

2.The innovation of this paper is noteworthy. This paper innovatively proposes to estimate the probability of examples being incorrect and applies instance-dependent thresholds based on these estimates. This approach is more nuanced and potentially more effective as it takes into account the individual characteristics of each example. It is the first to propose the estimation of instance-dependent thresholds in SSL. This novel approach represents could open up new avenues for research in the field.

3.The authors also provide sufficient theoretical analysis about their proposed method. They present a theorem that shows that for samples that satisfy their instance-dependent threshold function, the likelihood of the pseudo-labels being correct is lower-bounded. This provides a solid foundation for the proposed method and helps to convince readers of its assumption and correctness.

**Weaknesses:**

1.Certain aspects of the paper could benefit from further explanation and clarification. Specifically, the relationship between the Quality-Quantity Trade-off and the effective dynamic value is not clearly articulated in the main body of the paper. These are key components of the proposed method, and their interaction could significantly impact the performance of the method.

2.The paper does not sufficiently discuss the limitations of the proposed method. Every method has its limitations and potential drawbacks, and a thorough discussion of these is crucial for a balanced and comprehensive presentation of the work.

3.The empirical improvement of the proposed method is marginal when the amount of labeled data increases. This suggests that the method's performance may not scale well with larger labeled datasets.

4.The experimental evaluation of the method is based on only three datasets. This limited number of datasets may not provide a comprehensive evaluation of the method's performance.

**Questions:**

1.What might be the potential drawbacks or limitations of this method? How could these impact its applicability or performance in certain scenarios or with certain types of data?

2.The experimental evaluation of the method is based on only three datasets. Could the authors explain their choice of datasets and how representative these datasets are of the types of data the method would encounter in real-world applications? Would the performance of the method vary significantly if tested on other datasets?

3.The authors may need to clarify the relationship between the Quality-Quantity Trade-off and the effective dynamic value. How do these two factors interact within the context of their proposed method?

4.The paper discusses the balance between the quality and quantity of pseudo-labels. Could the authors provide some intuition of how this balance can be achieved in practice?

5.This paper is highly related to the problem of learning with instance-dependent label noise. To clearly demonstrate the effectiveness, can authors provide a comparison with methods for handling instance-dependent label noise?

**Limitations:**

The main paper seems to lack a comprehensive discussion on the limitations of the proposed method. It would be beneficial for the authors to conduct a thorough review of their method to identify any potential limitations. I suggest including a separate section in the main paper dedicated to discussing these limitations, which would provide a more balanced and complete perspective of the proposed method.

---

> ### Author Rebuttal · Authors · 2023-08-10
>
> Thank you for acknowledging our contributions and posing a wide range of valuable questions & suggestions, all of which have been very inspiring to us. We sincerely hope that our response can address your questions and concerns:
>
> > **W1 & Q3: Relationship between Quality-Quantity trade-off and efficitive dynamic thresholds.**
>
> We apologize for the lack of clarity in our main paper, and you're absolutely correct; this concept holds significant importance. Let's begin by considering the correlation between dynamic thresholds and the quality-quantity trade-off. Dynamic thresholds have now emerged as the primary solution for quality-quantity trade-off. In the initial training stages, an excessively high fixed threshold leads to an excessive filtration of samples, thereby compromising quantity. Conversely, during the later training stages, a fixed threshold set too low fails to effectively filter any samples, thereby compromising quality.
>
> In addition, to better understand and set restraint on the dynamic threshold, we define an "effective dynamic threshold" (Assumption 2), whose left-hand side can be viewed as the gain of the quality of the pseudo-label after introducing $\kappa$, and the right-hand side can be view as the loss of the quantity of the pseudo-label after introducing $\kappa$. So we're assuming that, with an effective dynamic threshold, we expect the increase in the quality of the pseudo-label must at least match the loss of the quantity of the pseudo-label.
>
> > **W2 & Q1: Didn't discuss the limition of InstanT & When InstanT could fail.**
>
> Thank you for this constructive suggestion, here we will discuss some of the potential limitations of InstanT, and we will include them in the official version of our paper.
>
>  - When there is minimal label error generated by the classifier, InstanT has no significant differences with other SSL methods. The instance-dependent thresholds generated by InstanT becomes trivial simply and reduce to $\kappa$.
>  - InstanT is subjective to the influence of transition matrix estimation, if, for some reason, the estimation of the transition matrix is extremely poor, this could potentially hinder the thresholding of InstanT.
>
> Therefore, InstanT could potentially obtain no significant improvements on simpler datasets and with abundant labeled sets, for instance, on the SVHN dataset, or on the CIFAR-10 dataset with abundant labeled samples.
>
> > **W3: Performance of InstanT on larger labeled datasets.**
>
> Thank you for raising this concern, this has been a commonly mentioned issue. We would like to point out that, since learning on larger labeled datasets is relatively easy, the performance limit on larger labeled datasets has almost been fully exploited, and converged towards the fully-supervised results. This means the performance gap will almost always be marginal on larger labeled datasets, similar results are verified in recent SSL papers as well [1,2].
>
> > **W4 & Q2: Number of datasets is limited.**
>
> Thank you for the suggestion, the datasets we selected are the most commonly used benchmarks in SSL with an adequate level of challengingness, all of which are widely acknowledged datasets to simulate real-world problems. In addition, to fully address your concern, we also conduct experiments on ImageNet-100, with 100 labeled samples per class, to evaluate our method. All results are obtained from a ResNet-50 train from scratch for 500,000 iterations on ImageNet-100 using random seed {0}.
>
> | Methods | Top-1 Accuracy | F-1 Score |
> | ------ | ------ | ------ |
> | FixMatch | 0.6624 | 0.6559 |
> | AdaMatch | 0.6860 | 0.6822 |
> | FreeMatch | 0.6578 | 0.6529 |
> | InstanT | **0.6994** | **0.6972** |
>
> Table 3-1
>
> As suggested in Table 3-1, InstanT also showcases strong performances on large-scale real-world datasets such as ImageNet.
>
> > **Q4: Intuition on how InstanT better balances between quantity-quality trade-off.**
>
> Basically, existing methods attempt to find a better balance between quantity and quality trade-offs by assigning dynamic thresholds that are dependent on the training progress. InstanT further improves this concept by utilizing the simple intuition that "some unlabeled samples are more likely to have incorrect pseudo-labels than others". Once we can determine the probability of which sample is more likely to have an incorrect pseudo-label, InstanT will adjust its label assignment threshold based on such likelihood, hence increasing the threshold level for samples with a larger noisy probability.
>
> > **Q5: Comparsion on methods handling instance-dependent label noise.**
>
> Here we will try to implement PTD [3] to test its performance in SSL. Our implementation strategy is for PTD to leverage labeled samples as anchor points to learn the transition matrix. We also apply two different settings for a fair comparison, where PTD will adapt fixed and dynamic thresholds to choose pseudo-label (noisy label) respectively. The version where PTD uses a dynamic threshold will be referred to as PTD(DT). Since PTD requires the number of anchor points to be larger than $C\times M$, where $C$ is the number of classes and $M$ is the number of parts (hyper-para), labels per class must be larger, hence we only evaluate on CIFAR-10(250).
>
> | Methods | CIFAR-10(250) |
> | ------ | ------ |
> | PTD | 0.9723±0.04 |
> | PTD(DT) | 0.9711±0.01 |
> | InstanT | **0.9808±0.00** |
>
> From the results, we can observe that PTD also achieves good results, but cannot be used for cases when the number of labels per class is limited. The main difference between PTD(DT) and InstanT is their thresholding method, the better performance of InstanT further verifies the effectiveness of the instance-dependent thresholds.
>
> Full reference info will be omit due to character limit
>
> [1] Freematch: Self-adaptive thresholding for semi-supervised learning
>
> [2] Softmatch: Addressing the quantity-quality trade-off in semi-supervised learning
>
> [3] Part-dependent label noise: Towards instance-dependent label noise

---

> > ### Comment · Reviewer_Cia8 · 2023-08-15
> >
> > Thank you for your reply. The response addresses my concerns well. I keep my score on acceptance.

---

> > > ### Author Response · Authors · 2023-08-17
> > > **Thanks for responding to our rebuttal!**
> > >
> > > Dear reviewer Cia8, we are happy to know that we have addressed your concerns well, and we're truly grateful for your kind praises to our work. We're committed to continuously refining our paper, thus we welcome any new insights and suggestions that you may still have during the rolling discussion phase.

---

### Official Review · Reviewer_zzrr · 2023-07-05

**Soundness:** 3 good
**Presentation:** 4 excellent
**Contribution:** 3 good
**Rating:** 6
**Confidence:** 3

**Summary:**

This paper proposes a semi-supervised method with instance-dependent thresholds (InstanT), which can assign different thresholds to individual unlabeled data based on the instance-dependent label noise level and prediction confidence. Also, this paper provides a theoretical analysis of the proposed InstanT. Extensive experiments have shown the effectiveness of the proposed method.

**Strengths:**

1. This paper proposes an Instance-dependent Thresholding strategy for semi-supervised learning. Also, the proposed method can vouch for the reliability of pseudo-labels it assigns with a theoretical guarantee.

2. This paper is easy to read and well organized.

3. The experimental results and the proof of  Theorems in this paper seem to be solid.


**Weaknesses:**

1. This paper is devoted to addressing the SSL problem with dynamic thresholding. However, there is a lack of some SOTA methods to be compared with the proposed method. For example,
[1] Guo et al., Class-imbalanced semi-supervised learning with adaptive thresholding, ICML, 2022.
[2] Yang et al., Class-aware contrastive semi-supervised learning, CVPR, 2022.
[3] Wang et al., Freematch: Self-adaptive thresholding for semi-supervised learning, ICLR, 2023.
[4] Chen et al., Softmatch: Addressing the quantity-quality trade-off in semi-supervised learning, ICLR, 2023.

2. Most previous SSL methods have reported their performance on both datasets: SVHN and ImageNet. It is therefore suggested that the authors supplement some experiments to evaluate the generalization performance of the proposed method.

3. It is suggested to omit some symbols whose meanings have been explained in the previous context, so as to increase the brevity of this paper.

4. The proposed method is motivated by the learning with noisy labels. Therefore, it is necessary to review some related methods to give readers a whole picture.


**Questions:**

Please refer to the Weaknesses.

**Limitations:**

The limitations of this paper should be discussed.

---

> ### Author Rebuttal · Authors · 2023-08-10
>
> We appreciate your invaluable feedbacks and suggestions, all of which will significantly contribute to the enhancement of our work. Please find our response addressing your concerns:
>
> >  W1: Lack of comparsion with SOTA (FreeMatch, SoftMatch etc.)
>
> Thank you for raising this question, we have conducted comprehensive evaluations against the two most recent SOTA methods - FreeMatch [1] and SoftMatch [2], as you suggested.
>
> First, we present the results under conventional settings [1,2], where all results are trained from scratch with WRN28-2, we fix all methods with random seed "0".
>
> | Methods | CIFAR-10(10) | CIFAR-10(40) | CIFAR-10(250) | CIFAR-100(400) | CIFAR-100(2500) |
> | ------ | ------ | ------ | ------ | ------ | ------ |
> | SoftMatch | 0.7557 | 0.9464 | 0.9517 | 0.5057 | 0.6622 |
> | FreeMatch | 0.9193 | **0.9512** | 0.9506 | 0.4920 | 0.6659 |
> | InstanT | **0.9250** | 0.9510 | **0.9525** | **0.5217** | **0.6709** |
>
> Table 2-1
>
> Under conventional settings, we can observe that InstanT showcases strong performance against SOTA baselines, especially on the most challenging cases,e.g. CIFAR-10(10) and CIFAR-100(400), indicating when there exist large label errors, InstanT can more effectively filter them out.
>
> We also present the results with pre-trained ViT, settings are aligned with Table 1 from our paper.
>
> | Methods | CIFAR-10(10) | CIFAR-10(40) | CIFAR-10(250) |  CIFAR-100(200) | CIFAR-100(400) | STL-10(10) | STL-10(40) |
> | ------ | ------ | ------ | ------ | ------ | ------ | ------ | ------ |
> | SoftMatch | 0.8003±0.09 | 0.9795±0.01 | 0.9814±0.00 | 0.7057±0.01 | 0.7821±0.01 | 0.6510±0.09 | 0.8370±0.04 |
> | FreeMatch | 0.7721±0.05 | **0.9811±0.00** | **0.9819±0.00** |  **0.7629±0.02** | **0.7938±0.00** |  0.6230±0.14 | 0.8496±0.03 |
> | InstanT | **0.8732±0.10** | 0.9793±0.00 | 0.9808±0.01 | 0.7417±0.00 | 0.7880±0.00 | **0.6939±0.07** | **0.8509±0.03** |
>
> Table 2-2
>
> From Table 2-2, we can observe that, while InstanT obtained the best results in 3 out of 7 cases, it's average improvement is much more significant than FreeMatch, and for cases where its performance is worse, the gaps are relatively marginal.
>
> > **W2: Lack of results on SVHN and ImageNet.**
>
> Thank you for raising this concern, we have added more experiments on ImageNet-100, with 100 labeled samples per-class, to evaluate our method against selected SOTA baselines. All results are obtained using  a ResNet-50 trained for 500,000 iterations, with random seed {0}.
>
> | Methods | Top-1 Acc. | F-1 Score |
> | ------ | ------ | ------ |
> | FixMatch | 0.6624 | 0.6559 |
> | AdaMatch | 0.6860 | 0.6822 |
> | FreeMatch | 0.6578 | 0.6529 |
> | InstanT | **0.6994** | **0.6972** |
>
> Table 2-3
>
> As we can observe from Table 2-3, InstanT shows strong generalization capability and scalability on large benchmarks such as ImageNet-100, surpassing a range of SOTA baseline methods.
>
> As for the SVHN dataset, since it is considered a "simpler" case, running InstanT on SVHN cannot effectively differentiate it from other methods. Therefore, given limited time and computational resources, we forfeit to display the results here. Alternatively, if the reviewer still deems it necessary, we could also showcase these results during the rolling discussion phase.
>
>  > **W3: Omit previously defined symbols.**
>
> We thank the reviewer for posing this feedback. Indeed, for brevity, we will avoid over-defining symbols in the official version of our paper.
>
> > **W4: Lack of review for label noise papers.**
>
> Here we will give a concise overview of the relevant label noise learning papers, a more comprehensive review will be added to the main paper at a later stage.
>
> The most relevant line of work is the modeling of label noise. The pioneering work introduced the concept of learning with class-dependent label noise, which has been widely recognized [3,4,5]. Noteworthy strategies in this area include anchor-point estimation [6], end-to-end estimation [7], mixture proportion estimation [3], etc.
>
> Meanwhile, another line of research has concentrated on tackling instance-dependent label noise [8], which reflects a more realistic scenario. In this regard, notable works include the part-dependent anchor-points [9] and modeling through DNNs [10].
>
> Once again, we would like to thank the reviewer for spending your valuable time reviewing our paper, and we welcome any new questions & suggestions the reviewer might have after rebuttal.
>
>  - [1] Wang, Yidong, et al. "Freematch: Self-adaptive thresholding for semi-supervised learning." arXiv preprint arXiv:2205.07246 (2022).
>  - [2] Chen, Hao, et al. "Softmatch: Addressing the quantity-quality trade-off in semi-supervised learning." arXiv preprint arXiv:2301.10921 (2023).
>  - [3] Scott, Clayton, Gilles Blanchard, and Gregory Handy. "Classification with asymmetric label noise: Consistency and maximal denoising." Conference on learning theory. PMLR, 2013.
>  - [4] Natarajan, Nagarajan, et al. "Learning with noisy labels." Advances in neural information processing systems 26 (2013).
>  - [5] Patrini, Giorgio, et al. "Making deep neural networks robust to label noise: A loss correction approach." Proceedings of the IEEE conference on computer vision and pattern recognition. 2017.
>  - [6] Liu, Tongliang, and Dacheng Tao. "Classification with noisy labels by importance reweighting." IEEE Transactions on pattern analysis and machine intelligence 38.3 (2015): 447-461.
>  - [7] Li, Xuefeng, et al. "Provably end-to-end label-noise learning without anchor points." International conference on machine learning. PMLR, 2021.
>  - [8] Cheng, Jiacheng, et al. "Learning with bounded instance and label-dependent label noise." International conference on machine learning. PMLR, 2020.
>  - [9] Xia, Xiaobo, et al. "Part-dependent label noise: Towards instance-dependent label noise." Advances in Neural Information Processing Systems 33 (2020): 7597-7610.
>  - [10] Yang, Shuo, et al. "Estimating instance-dependent label-noise transition matrix using dnns." (2021).

---

> > ### Comment · Reviewer_zzrr · 2023-08-14
> > **Thanks for your detailed response.**
> >
> > Thank you for carefully responding to my comments. I have read your rebuttals to all reviews. Overall, I am convinced that this will be a valuable contribution to NeurIPS 2023 and I will stay with my original rating of Weak Accept.

---

> > > ### Author Response · Authors · 2023-08-14
> > > **Thank you for responding to our rebuttal!**
> > >
> > > Dear reviewer zzrr, we sincerely thank you for reading through all of our rebuttals, we are strongly encouraged by your acknowledgments of our contributions! We're committed to continuously refining our paper, thus we welcome any new insights and suggestions that you may still have during the rolling discussion phase.

---

### Official Review · Reviewer_1e9N · 2023-07-20

**Soundness:** 3 good
**Presentation:** 3 good
**Contribution:** 2 fair
**Rating:** 5
**Confidence:** 4

**Summary:**

- Assumption for instance-dependent threshold setting and theoretical proof based on it
- Transition matrix modeling (estimator design) to reduce label error through instance-dependent threshold function
- This paper shows good performance on various datasets (CIFAR10, 100, STL-10)

**Strengths:**

1) A new approach in SSL called instance-dependent threshold setting
2) Solid theoretical proof
3) Excellent performance, especially in environments with little labeled data

**Weaknesses:**

1) Transition matrix modeling for threshold calculation for each instance is required (complexity is expected to increase).
2) There is no consideration for class imbalance, which is mainly dealt with in SSL. If there is a class imbalance problem, it may not be a good idea to set different thresholds for each instance.
3) Comparative papers (FlexMatch, Dash, AdaMatch, etc.) are papers published before 2022. Comparisons with recent papers have not been made. In addition, AdaMatch, which is mainly compared, is a paper dealing with domain adaptation issues.

**Questions:**

1) The running time in Table 2 was 1s lower than Fix and AdaMatch. Do you have the result in terms of the amount of calculations and the number of parameters?
2) If the reason for applying Distribution Alignment in InstanT-II in Table 3 is to eliminate class imbance, what is the reason why performance improves when DA is applied in ablation study? (The experimental environment in the paper appears to be a situation with low imbalance using the same number of labeled data for each class).
3) Usually SSL papers show error rates of CIFAR10/100. How does the error rate come out?
4) What about the results on larger sized images (e.g. ImageNet)?
5) Papers on instance dependent pseudo-labeling already exist. Is there any difference from that? For example, the paper SimMatch: Semi-supervised Learning with Similarity Matching (2022 CVPR) also uses instance dependent similarity. This paper theoretically proves the validity of using instance-level threshold, but is there any excellence or novelty compared to the similarity method used in the above paper?
6) The results of the comparative papers presented in the proposed paper and the results of the papers below are different. Is it an accurate comparison? Is there a reason why the second paper below is not included in the result table?
- SimMatch: Semi-supervised Learning with Similarity Matching (2022 CVPR)
- SOFTMATCH: ADDRESSING THE QUANTITY-QUALITY TRADE-OFF IN SEMI-SUPERVISED LEARNING (2023 ICLR)
7) Please also respond to the weaknesses pointed out.

**Limitations:**

The authors have adequately addressed the limitations and potential negative societal impact of their work.

---

> ### Author Rebuttal · Authors · 2023-08-10
>
> Dear reviewer 1e9N, we are grateful for your comprehensive review and the substantial range of questions you have raised.
>
> > **W1: Computational complexity**
>
> As suggested in Table 2 of our paper, InstanT brings a minimal increase in terms of training time. Here we present you with more run-time analysis from other datasets, all speeds are measured with Nvidia RTX 4090 GPUs.
>
> Pre-trained results with ViT:
>
> | Methods | CIFAR-10 | CIFAR-100 |
> | ------ | ------ | ------ |
> | FixMatch | 104.4 | 100.9 |
> | AdaMatch | 104.6 | 101.4 |
> | InstanT | 106.7 | 104.2 |
>
> Table 1-1: Seconds per-epoch during the training of each method.
>
> Overall, while the complexity of InstanT indeed increases, the training speed is not significantly impaired compared with other baseline methods. More importantly, we believe that the code can be further optimized and reduce the run-time.
>
> > **W2: Performance of InstanT when there exists class-imbalance**
>
> Thank you for posing this concern when there exists class imbalance, InstanT is still expected to maintain a robust performance. We will address this from two perspectives: (1) Theoretical capabilities of InstanT with motivating examples; (2) Empirical performance of InstanT under imbalanced scenarios.
>
>  - In the presence of class imbalance, this disparity typically manifests in the class posterior (prediction); InstanT is sensitive to the distribution and can leverage the labeled samples to probe such imbalanced class posterior, and attain an estimation to account for the probability of being misclassified.
>
>  - Lastly, to fully address your concern, we also conducted comprehensive experiments on class-imbalance cases to empirically verify the performance of InstanT. All results are trained with Wide-ResNet-28-2 for $2^{18}$ iteration, following commonly used settings [4,5], we fix all methods with random seed {0}, imbalance ratio $\gamma$ = {50,100,150}
>
> | Methods | $\gamma$=50 | $\gamma$=100  | $\gamma$=150 |
> | ------ | ------ | ------ | ------ |
> | Dash | 0.7828 | 0.7077 | 0.6572 |
> | FlexMatch | 0.7977 | 0.7069 | 0.6433 |
> | AdaMatch | 0.7937 | 0.7278 | 0.6571 |
> | FreeMatch | 0.7971 | 0.7208 | 0.6354 |
> | InstanT | **0.7993** | **0.7346** | **0.6797** |
>
> Table 1-2: CIFAR-10, $N_1$=500, $M_1$=4000.
>
> | Methods | $\gamma$=50 | $\gamma$=100  | $\gamma$=150 |
> | ------ | ------ | ------ | ------ |
> | Dash | 0.8118 | 0.7528 | 0.6940 |
> | FlexMatch | 0.8117 | 0.7445 | 0.6947 |
> | AdaMatch | 0.8198 | 0.7496 | 0.6969 |
> | FreeMatch | 0.8196 | 0.7543 | 0.7001 |
> | InstanT | **0.8216** | **0.7602** | **0.7116** |
>
> Table 1-3: CIFAR-10, $N_1$=1500, $M_1$=3000.
>
> The above results verified our justification, under class-imbalanced scenarios, the instance-dependent threshold still exhibited outstanding performances compared with other baseline methods. As the class-imbalance ratio increases, we can observe a more significant advantage of InstanT. This corroborates our hypothesis - when classes are highly imbalanced, InstanT can indeed filter out substantial label errors and maintain a much more robust performance.
>
> > **W3: Lack of comparsion with recent methods & AdaMatch as a domain adapation method**
>
> Due to character limit, please find our response to this question at the General Response 1 at the top.
>
> It's correct that AdaMatch can be used for Domain Adaptation (DA), but it's a SSL method as well. Due to the inherent similarity, DA is usually considered a highly relevant topic to SSL. That's why AdaMatch is named "A Unified Approach to Semi-Supervised Learning and Domain Adaptation".
>
> > **Q1: Number of operations and number of parameters**
>
> Due to the character limit, please find our response to this question in the general response section above.
>
> > **Q2: Purpose of Distribution Alignment in class-balance case**
>
> Distribution Alignment is not only useful when the initial labeled set is imbalanced. A more common issue in SSL is the self-generative class-imbalance, meaning during the process of psudeo-labeling, the error of classifiers will acculumate and eventually leading to the domainace of certain psudeo-label class. Distribution alignemnt is therefore employed to modulate the predictions made by the classifier and alleviates the issue of imbalanced pseudo-labels.
>
> > **Q3: Results in error rates**
>
> The error rate is simply 1 - classification accuracy.
>
> > **Q4: Results on ImageNet**
>
> Due to the character limit, please find our response to this question at the General Response 2 at the top.
>
> > **Q5: Differences between SimMatch.**
>
> **SimMatch did not introduce instance-dependent threshold, it applies a fixed threshold to all samples.** It's core contribution lies in an enhanced form of consistency regularization, considering instance-level similarity. SimMatch essentially refines the application of consistency regularization to prevent overfitting. Whereas InstanT focus on devising a novel thresholding approach aimed at enhancing the filtration of label errors more efficiently.
>
> > **Q6: Reported results are different from the original papers, lack of coparsion with SoftMatch**
>
> Thanks for the suggestion, we have added the comparsions with SoftMatch in the Table 0-1 an 0-2 of the General Response.
>
> The discrepancy between the results from Table 1 and the original papers is because we have adapted a new trending setting in SSL [3], where we use pre-trained ViT as backbone instead of WRN training from scratch. This new setting is more computationally efficient, and all baseline methods have been tuned to their favorable hyper-parameters free from bias [3].
>
> We also included results under conventional settings, those results can be found in the Appendix, as well as Table 0-1 from General Response 1. Where you will find that the results are overall consistent with the results reported in other papers.
>
> Due to character limits, please find the reference list at the end of General Response.

---

> > ### Comment · Reviewer_1e9N · 2023-08-11
> >
> > I agree that the authors have sufficiently responded to my comments. Given that, currently I'm willing to change my score.
> >
> > Best Regards

---

> > > ### Author Response · Authors · 2023-08-11
> > > **Thank you for responding to our rebuttal!**
> > >
> > > Dear reviewer 1e9N, we sincerely thank you for reviewing our rebuttal diligently, your acknowledgment means a lot to us. We're committed to continuously refining our paper, thus we welcome any new insights and suggestions that you may still have during the rolling discussion phase.

---

### Author Rebuttal · Authors · 2023-08-10

Dear reviewers, please find our general responses to some of the commonly asked questions, due to character limits in some of the individual rebuttal, we kindly refer you to see our response here:

> **GA1: Comparsions with more recent baselines**

We have included the comparsions with two recent baselines, FreeMatch (ICLR'23) and SoftMatch (ICLR'23). First, we present the results under conventional settings [1,2], where all results are trained from the scratch with WRN28-2, we fix all methods with random seed "0".

| Methods | CIFAR-10(10) | CIFAR-10(40) | CIFAR-10(250) | CIFAR-100(400) | CIFAR-100(2500) |
| ------ | ------ | ------ | ------ | ------ | ------ |
| SoftMatch | 0.7557 | 0.9464 | 0.9517 | 0.5057 | 0.6622 |
| FreeMatch | 0.9193 | **0.9512** | 0.9506 | 0.4920 | 0.6659 |
| InstanT | **0.9250** | 0.9510 | **0.9525** | **0.5217** | **0.6709** |

Table 0-1

Under conventional settings, we can observe that InstanT showcases strong performance against SOTA baselines, especially on hard cases (e.g. CIFAR-10(10) and CIFAR-100(400)), indicating when there exists large label errors, InstanT can more efficitively filter them out.

We also present the results from pre-trained ViT, settings are aligned with Table 1 from our paper.

pre-trained results
| Methods | CIFAR-10(10) | CIFAR-10(40) | CIFAR-10(250) |  CIFAR-100(200) | CIFAR-100(400) | STL-10(10) | STL-10(40) |
| ------ | ------ | ------ | ------ | ------ | ------ | ------ | ------ |
| SoftMatch | 0.8003±0.09 | 0.9795±0.01 | 0.9814±0.00 | 0.7057±0.01 | 0.7821±0.01 | 0.6510±0.09 | 0.8370±0.04 |
| FreeMatch | 0.7721±0.05 | **0.9811±0.00** | **0.9819±0.00** |  **0.7629±0.02** | **0.7938±0.00** | 0.6230±0.14 | 0.8496±0.03 |
| InstanT | **0.8732±0.10** | 0.9793±0.00 | 0.9808±0.01 | 0.7417±0.00 | 0.7880±0.00 | **0.6939±0.07** | **0.8509±0.03** |

Table 0-2

From Table 0-2, it is evident that eventhough InstanT only achieved the highest scores in 3 out of 7 cases, its average enhancement is considerably more pronounced than that of FreeMatch. Furthermore, in situations where InstanT's performance shows a decline, the gaps are relatively marginal.

> **GA2: Experiments on large-scale datasets (ImageNet)**

We have added the experiments on ImageNet-100, with 100 labeled samples per-class, to evaluate our method against selected baseline methods. All results are obtained with a ResNet-50 trained for 500,000 iterations, with random seed {0}.

| Methods | Top-1 Accuracy | F-1 Score |
| ------ | ------ | ------ |
| FixMatch | 0.6624 | 0.6559 |
| AdaMatch | 0.6860 | 0.6822 |
| FreeMatch | 0.6578 | 0.6529 |
| InstanT | **0.6994** | **0.6972** |

Table 0-3

As we can observe from Table 0-3, InstanT showcases strong performances on large-scale real-world datasets as well.

References:

- [1] Freematch: Self-adaptive thresholding for semi-supervised learning
- [2] Softmatch: Addressing the quantity-quality trade-off in semi-supervised learning
- [3] Usb: A unified semi-supervised learning benchmark for classification
- [4] Class-imbalanced semi-supervised learning with adaptive thresholding
- [5] Crest: A class-rebalancing self-training framework for imbalanced semi-supervised learning

---

### Decision · Program_Chairs · 2023-09-21

**Decision:**

Accept (poster)

**Comment:**

This work investigates the instance-dependent thresholds for semi-supervised learning. It is conducted in a principled manner and the experimental study demonstrates its potential. Reviewers raised a number of issues on this work. The authors provided feedback and this helped to address the concerns. After the author-reviewer discussion phase, all scores converge to leaning to accept this work. AC also read the submission, review and response and discussed this work with SAC. In conclusion, this work has its value and significance in addressing the instance-dependent thresholding issue by taking a theoretical perspective. The overall quality of this work meets the requirement of this conference. Acceptance is therefore recommended. Meanwhile, this work needs to adequately consider the review comments when preparing the final version of this submission. In particular, this work shall improve the clarity at multiple places, especially those related to theoretical analysis, and more rigorously present Assumption 2.